# Synergetic Effect of Dual Functional Monomers in Molecularly Imprinted Polymer Preparation for Selective Solid Phase Extraction of Ciprofloxacin

**DOI:** 10.3390/polym13162788

**Published:** 2021-08-19

**Authors:** Ut Dong Thach, Hong Hanh Nguyen Thi, Tuan Dung Pham, Hong Dao Mai, Tran-Thi Nhu-Trang

**Affiliations:** 1Department of Polymer Chemistry, Graduate University of Science and Technology, Vietnam Academy of Science and Technology, Ha Noi 100000, Vietnam; 2Faculty of Pharmacy, Ton Duc Thang University, Ho Chi Minh City 700000, Vietnam; 3Faculty of Chemistry, University of Science, Vietnam National University, Ho Chi Minh City 700000, Vietnam; nthhanh2397@gmail.com (H.H.N.T.); t.dung597@gmail.com (T.D.P.); hongdao4567@gmail.com (H.D.M.); 4Faculty of Environmental and Food Engineering, Nguyen Tat Thanh University (NTTU), Ho Chi Minh City 700000, Vietnam

**Keywords:** ciprofloxacin, imprinted polymer, dual functional monomer, solid phase extraction

## Abstract

Background: Ciprofloxacin (CIP), an important broad-spectrum fluoroquinolone antibiotic, was often used as a template molecule for the preparation of imprinted materials. In this study, methacrylic acid and 2-vinylpyridine were employed for the first time as dual functional monomers for synthesizing ciprofloxacin imprinted polymers. Methods: The chemical and physicochemical properties of synthesized polymers were characterized using Fourier transform-infrared spectroscopy, thermogravimetric analysis-differential scanning calorimetry, scanning electron microscopy, and nitrogen adsorption-desorption isotherm. The adsorption properties of ciprofloxacin onto synthesized polymers were determined by batch experiments. The extraction performances were studied using the solid phase extraction and HPLC-UV method. Results: The molecularly imprinted polymer synthesized with dual functional monomers showed a higher adsorption capacity and selectivity toward the template molecule. The adsorbed amounts of ciprofloxacin onto the imprinted and non-imprinted polymer were 2.40 and 1.45 mg g^−1^, respectively. Furthermore, the imprinted polymers were employed as a selective adsorbent for the solid phase extraction of ciprofloxacin in aqueous solutions with the recovery of 105% and relative standard deviation of 7.9%. This work provides an alternative approach for designing a new adsorbent with high adsorption capacity and good extraction performance for highly polar template molecules.

## 1. Introduction

Ciprofloxacin (CIP) is an important antibiotic belonging to the class of fluoroquinolones. It is a broad spectrum of an antibacterial agent widely applied to treat various infectious diseases in animals and humans [1]. It is reported that only 30% of CIP can be metabolized inside the body and a large amount of CIP has been emitted into the environment [1]. Therefore, the intense use of CIP worldwide causes potential threats such as bacterial resistance, allergy, and toxicity [2,3]. For the protection of public health, many countries have established different maximum residue limits for CIP in various food samples. For example, CIP’s tolerance is 0.1 mg L^−1^ in milk and 0.5 mg kg^−1^ (total of CIP and enrofloxacin) in porcine liver [4].

Many analytical methods have been developed to determine CIP residue in various food and environmental samples, such as spectrophotometry [5,6,7], capillary electrophoresis [8], and high-performance liquid chromatography using UV [9], fluorescence [10,11], diode array [12], and mass spectrometry [13]. For the instrumental analysis, sample pre-treatment is essential for the purification and enrichment of trace CIP residue in complex matrices. Several separation methods have been employed to extract, purify, and preconcentrate CIP from real samples, including solid phase extraction (SPE) [3], stir bar sorptive extraction [14], magnetic nanoparticles SPE [14], liquid–liquid extraction [15], dispersive liquid–liquid microextraction [14], and immunoaffinity chromatography [14]. Overall, SPE is widely employed as a purification and preconcentration method due to its performance and small organic solvent volumes used [16]. Commercial SPE cartridges are available with different adsorbents, such as C_8_, C_18_, Al_2_O_3_, silica, and resin. However, these sorbents are non-recyclable, non-selective, and easily extract several compounds, including the analyte and the interferences. Thus, it is important to explore new adsorbents for CIP extraction with excellent selectivity, durability, and reusability.

Molecularly imprinted polymers (MIP) are highly selective adsorbents prepared by the copolymerization of suitable functional monomers and crosslinkers with the presence of a target template molecule [17]. These imprinted polymers have attracted considerable research attention due to their interesting properties such as reproductivity, low cost, ease of preparation, and high selectivity toward target molecules [18]. Due to their unique properties, MIPs have been widely employed in various applications such as chemical sensors [19], biomimetic catalyst [20], drug delivery [21], protein crystallization [22], chromatography [23], bioanalysis [24], and solid phase extraction [1].

A CIP-imprinted polymer (CIP-MIP) was synthesized for the first time in 2006 [25]. Then, many studies have reported CIP-MIP synthesis by using different functional monomers, such as methacrylic acid [1,25,26,27,28,29,30,31,32], 2-vinylpyridine [32], 1-vinyl-3-ethylimidazolium bromide [33], and 4-vinylbenzoic acid [3]. CIP imprinted materials were applied for adsorption [34] or detection of trace CIP residues in various biological and environmental samples using chemosensor and biosensor [35], electrochemical sensor [36,37], and optosensor [38] techniques. Furthermore, Tarly et al. proposed using semi-empirical quantum chemistry stimulation for synthesizing CIP-MIP. Both the computational calculation and the practical experiments showed that MIP synthesized with acrylamide monomer exhibited the highest specific selectivity factor and adsorption capacity [39]. Moreover, MIPs were usually synthesized in the non-polar or weakly polar organic solvent and exhibited good recognition performance for non or weakly polar template molecules [40]. Hydrogen bonds are the driving force for forming specific biding sites for templates containing highly polar functional groups. These interactions could interfere with water molecules, resulting in a reduction in the imprinted polymer selectivity [1,31,41]. The competitive adsorption of water onto specific binding sites via hydrogen bond prevents interaction between the template molecules and imprinting site, which led to the formation of highly non-selective adsorption sites [1]. It is reported that the multifunctional monomer imprinting strategy provides multiple types of intermolecular forces with the template structure [42]. The imprinted polymers synthesized with dual functional monomers showed excellent molecular recognition and a high adsorption capacity, especially for templates containing strong polar functional groups [42,43,44,45]. In a similar approach, Wang et al. reported the synthesis of CIP-MIP by using methacrylic acid and 2-hydroxyethyl methacrylate as stimuli-recognition elements [46]. Zhu et al. proposed using 1-ally-3-vinylimidazolium chorine and 2-hydroxyethyl methacrylate as a component of bifunctional monomers [1]. These strategies allowed the obtained polymer to interact strongly with CIP molecules in an aqueous solution via hydrogen bonds, electrostatic, hydrophobic, and π-π stacking interactions. Thus, the multifunctional monomer strategy is an effective method for synthesizing imprinted materials, especially for templates with polar functional groups such as CIP. Various advanced imprinting techniques (on surface, co-precipitation, emulsion, and suspension polymerization) have been developed for preparing imprinted materials [47,48,49,50]. The conventional bulk imprinting is essential for preparing imprinted materials because a simple, rapid, and pure MIPs’ production can be produced without sophisticated instrumentation. The bulk is an appropriate form for the application in solid phase extraction both in academia [33,41] and industrial manufacture [51].

In this study, methacrylic acid and 2-vinylpyridine, which provide complementary structure, size, and chemical properties toward CIP molecules, were employed for the first time as dual functional monomers in MIP preparation using a bulk imprinting technique. The complementary interactions of functional monomers (MAA and 2-VP) with CIP via hydrogen bond, electrostatic, and π-π stacking are expected to greatly improve the adsorption selectivity of the imprinted polymers. To obtain the best performance of CIP-MIP, several preparation parameters, such as porogenic solvent (methanol/acetic acid, chloroform/methanol, and dichloromethane/methanol), template/monomer molar ratio, and methacrylic acid/2-vinylpyridine molar ratio, were investigated and optimized. The FT-IR, TGA-DSC, SEM, and nitrogen adsorption–desorption analyses were employed for the physicochemical characterization of the obtained polymers. The adsorption properties of CIP onto MIP and non-imprinted polymer (NIP) were determined using batch adsorption experiments. Finally, CIP-MIPs were employed as an adsorbent for the selective extraction of CIP in water.

## 2. Materials and Methods

### 2.1. Materials

Ciprofloxacin 98.0%, methacrylic acid (MAA) 99%, 2-vinylpyridine (2-VP) 97%, ethylene glycol dimethacrylate (EDGMA) 98%, and azobisisobutyrontrile (AIBN) 12% wt in acetone were purchased from Sigma-Aldrich, St. Louis, MO, USA. Acetonitrile, methanol, acetic acid, phosphoric acid, and triethylamine with HPLC grade were purchased from Merck, Darmstadt, HL, Germany. The standard stock of CIP (500 mg L^−1^) was prepared in methanol/acetic acid (9:1, *v*/*v*), and the working solutions were diluted from the stock solution with deionized water. The standard solutions were stored at 4 °C to be stable for one week. The phosphoric acid solution 0.05% (pH 3) was prepared by adding 3.0 mL of phosphoric acid 85% to 1000 mL of pure water and adjusting the pH to 3.0 by triethylamine.

### 2.2. Methods

#### 2.2.1. Preparation of CIP-MIPs

The CIP-MIPs were synthesized using the bulk polymerization method. Briefly, CIP (0.1 mmol), methacrylic acid (0.66 mmol), 2-vinylpyridine (0.33 mmol), and 12 mL of porogen CHCl_3_/MeOH (9:1, *v*/*v*) were added into a screw-capped glass bottle. The mixture was sonicated for 15 min to obtain the homogenous solution and kept overnight at 4 °C for the formation of complex pre-polymerization. Next, ethylene glycol dimethacrylate (5 mmol) and azobisisobutyronitrile (20 mg) were added to the mixture. The oxygen in the bottle was removed by argon for 10 min. The polymerization was activated at 60 °C for 24 h in the thermostatic water bath. After that, the obtained bulk polymer was crushed and ground to obtain polymer particles from 35 to 100 μm. The CIP template was eliminated by washing with methanol/acetic acid (9:1, *v*/*v*) in an ultrasonic bath until no CIP was detected using HPLC-UV method. Finally, the obtained polymer particles were dried at 110 °C for 6 h. The corresponding non-imprinted polymer (NIP) was also prepared as a similar protocol but without the CIP template. The detailed MIP preparation parameters were summarized in Table 1.

#### 2.2.2. Polymer Characterization

The FT-IR spectroscopy analyses were conducted using an ATR-FTIR FT/IR6600A spectrometer, Seri A012761790, JASCO, Tokyo, Japan. Thermogravimetric analysis (TGA) and differential scanning calorimetry (DSC) were carried out using a TG-DSC LabSys Evo 1600, SETARAM, Austin, TX, USA. The samples were heated from 25 to 800 °C at a heat rate of 10 °C min^−1^ under nitrogen atmosphere. The nitrogen adsorption–desorption isotherms were conducted at 77 K using a Micromeritic tristar, Norcross, GA, USA. The polymers were outgassed at 100 °C for 8 h at 0.1 mbar before the analysis. The specific surface area was calculated using the Brunauer–Emmett–Teller (BET) model, using a cross-sectional area of 0.162 nm^2^ per nitrogen molecule. Scanning electron microscopy (SEM) was recorded using SEM S-4800, 10 kV, 7.9 mm, Hitachi, Tokyo, Japan.

#### 2.2.3. High Performance Liquid Chromatography Method

Chromatography analysis was conducted using HPLC equipment, Shimazu, Japan, with UV SPD-20A detector and a reversed-phase Water InertSustain AQ-C18 column (4.6 × 150 mm^2^, 3 μm particle size i.d.), Waters, USA. The mobile phase at pH 3 was applied in the isocratic mode with 82% of H_3_PO_4_ 0.05% and 18% of acetonitrile. The injection volume was set at 20 μL and 289 nm was employed as the detection wavelength. Retention times and areas of chromatographic peaks were used for qualitative and quantitative analyses, respectively.

#### 2.2.4. Adsorption Study

The individual adsorption isotherms of CIP onto polymer were determined by batch experiments. This study was carried out by equilibrating 10 mg of polymer and 20 mL of CIP solution in 50-milliliter centrifuge tubes. The initial CIP concentrations were varied from 0.14 to 10.0 mg L^−1^. The centrifuge tubes were slowly shaken for 3 h at 25 °C. The supernatants were filtered out by 0.22-micrometer nylon Millipore filters. The concentrations of CIP in the supernatants were determined using HPLC-UV. The equilibrium adsorption capacity was calculated as follows:(1)$$Q_{ads}=\frac{V_{o}\left(C_{i}-C_{e}\right)}{m_{p}}$$
where *Q_ads_* (mg g^−1^) is the equilibrium adsorbed amount of CIP; *C_i_* and *C_e_* (mg L^−1^) are the initial and final concentrations of the CIP solution, respectively; *V_o_* (L) is the volume of the CIP solution; and *m_p_* (g) is the mass of the polymer.

#### 2.2.5. Solid Phase Extraction Study

The obtained MIPs (40 mg) were employed for the preparation of SPE cartridges. A volume of 5 mL of CIP solution in water (0.1 mg L^−1^) was loaded. The SPE cartridge was washed with 3 mL of deionized water and then eluted by 3 mL of methanol/acetic acid. The final solution volume was adjusted to 1.5 mL. The CIP concentration in eluting solutions were determined using HPLC-UV in order to evaluate the SPE performance through recovery values.

## 3. Results

### 3.1. Effect of Polymerization Conditions

#### 3.1.1. Porogenic Solvent

The porogenic solvent is one of the most critical factors in preparing an imprinted polymer and determines the intermolecular interaction between the functional monomer and the template. The influence of the porogenic solvent on the extraction recovery of an imprinted polymer is shown in Table 1 (entries 1–3). The imprinted polymer MIP3, synthesized in dichloromethane:methanol (9:1, *v*/*v*), showed a slight extraction recovery for CIP (9.7%), while the higher extraction recovery (57.7%) was obtained when the polymer was synthesized in higher polar aprotic solvent chloroform:methanol (9:1, *v*/*v*). However, the polymer synthesized in a polar protic solvent as methanol:acid acetic (9:1, *v*/*v*) showed a low extraction recovery (36.5%). These results indicated that the highly polar solvent (chloroform/methanol) was suitable for preparing CIP-MIP and the weakly polar solvent unfavored the interaction between the monomer and the template molecule. In contrast, highly polar protic solvents such as methanol and acetic acid exhibited an intensely competitive interaction with functional monomers. Therefore, the mixture chloroform:methanol (9:1, *v*/*v*) was chosen as the porogenic solvent for an imprinted polymer preparation.

#### 3.1.2. Template/Functional Monomer Molar Ratio

Template/functional monomer molar ratio determines the composition, affinity, rigidity, and polymerization rate of the monomer. Three different templates/functional monomer ratios (1:10; 1:15, and 1:20) were evaluated. As shown in Table 1, the recovery was statistically comparable with an increasing number of functional monomers. However, the high template/functional monomer molar ratio increased the non-selective site on polymer and, therefore, caused the loss of the recognition properties [1,33]. For this reason, the template/functional monomer molar ratio 1:10 was used in the imprinted polymer preparation.

#### 3.1.3. Methacrylic Acid/2-Vinylpyridine Molar Ratio

The mixture of methacrylic acid and 2-vinylpyridine monomers with different ratios, which provide a complementary intermolecular interaction toward CIP, was evaluated to adjust the affinity of the obtained polymer. The MAA enhances the affinity and hydrophilicity of MIPs via hydrogen bonds, while 2-VP provides a hydrophobic and π-π stacking interaction with the template molecule. The synergetic effect of these functional monomers is expected to improve the extraction performance of the imprinted polymers. Four MAA/2-VP molar ratios (0.5:0.5; 0.66:0.33; 0.7:0.3 and 0.8:0.2) were evaluated in this study. Higher extraction recoveries (63–105%) were observed when dual functional monomers were used to prepare the imprinted polymer in which the MIP7, synthesized with the MAA/2-VP molar ratios of 0.66:0.33, showed the highest recovery (105%). These results indicated that the extraction performance of MIP was considerably improved by combining MAA and 2-VP as dual functional monomers. The complementary interactions of the hydrogen bond and the π-π stacking of dual-functional monomers favored the access to the recognition cavity and improved the extraction performance of a synthesized imprinted polymer [52].

### 3.2. Characterization of Polymers

The chemical structure of the obtained polymers was analyzed using FT-IR spectroscopy. As showed in Figure 1, the imprinted polymer (MIP7) and their corresponding non-imprinted polymer (NIP7) demonstrated similar FT-IR spectra. The results suggested that these polymers contained similar chemical compositions. The characteristic of the C-H stretching absorption peak appeared at 2950 cm^−1^. The strong stretching vibration absorption of C = O in EDGMA and MAA was observed at 1716 cm^−1^ [33]. The characteristics absorption peaks of 2-VP appearing at 1637 and 1450 cm^−1^ were due to the C = N and C = C stretching vibration of the pyridine ring [53]. The strong peak at 1141 cm^−1^ was due to the C–O stretching vibration of EDGMA. The band at 755 cm^−1^ was characteristic for C-H an out-of-plan bending vibration [28]. These results indicated that the 2-VP, MAA, and EDGMA monomers were successfully copolymerized during the imprinted polymer preparation.

The thermogravimetric analysis-differential scanning calorimetry (TGA-DSC) method was used for analyzing the thermal properties of the polymers. As showed in Figure 2, the first weight loss (9.5%) observed at below 220 °C was due to the loss of the adsorbed water and residue solvent molecules. From 250 to 450 °C, the principal weight loss of 63.1% was characteristic for the thermal decomposition of the polymer. The polymer was completely decomposed at 470 °C. Similar results were observed for the corresponding non-imprinted polymer (NIP7). The weight loss (7.8%) below 220 °C was due to the desorption of water and residue solvent molecules. The thermal decomposition occurred at 250 °C with a major weight loss of 62.6%. The non-imprinted polymer was also completely decomposed at 470 °C. The corresponding DSC thermogram of the imprinted polymer (MIP7) showed two endothermic peaks at 97 and 183 °C, which were due to removing adsorbed water and entrapped solvent molecules. The prominent endothermic peak at 365 °C might be due to the loss of pyridine and CO_2_ molecules [54]. The final endothermic peak at 431 °C could be explained by the pyrolysis with carbonization of the backbone. A similar result was observed for the DSC thermogram of the non-imprinted polymer (NIP7).

The textural properties of the synthesized polymers were determined using a nitrogen sorption isotherm. The results of the textural properties analyzed using a nitrogen adsorption–desorption isotherm showed that the synthesized polymers were a non-porous material (as shown in Figure 3). The imprinted polymer MIP7 had a specific surface area of 2.5 m^2^ g^−1^, while the non-imprinted NIP7 had a specific surface area of 5.4 m^2^ g^−1^. The morphology properties of polymers were confirmed using scanning electron microscopy (as shown in Figure 4). It is noted that the surface morphologies of the imprinted and non-imprinted polymer were similar. These results indicated that the bulk polymerization process in MIP preparation led to form non-porous and equant polymer particles.

### 3.3. Adsorption Properties

Batch adsorption experiments were used to determine the adsorption properties of CIP onto the as-prepared polymers. The selectivity of imprinted polymers was evaluated using the imprinting factor (IF = Q_MIP_/Q_NIP_) [1]. The adsorption isotherms of MIP2, MIP7, MIP10, and the corresponding non-imprinted polymers are shown in Figure 5.

When 2-VP was employed in MIP synthesis, the obtained polymers were non-water-compatible and weakly interacted with CIP molecules. The adsorption isotherms of MIP2 and NIP2 were S-type isotherms, characteristic of a low affinity of adsorbent material [55]. The maximum adsorbed amounts of CIP on MIP2 and NIP2 were 1.12 and 0.74 mg g^−1^, respectively, with the imprinting factor of 1.51. Thus, MIP2 was a selective adsorbent for CIP. These results indicated that the π-π stacking interaction between 2-VP and CIP could not be disturbed by methanol in the synthesis and water molecules in the rebidding experiments.

It was found that the imprinted polymer obtained from the dual functional monomers had a higher adsorption capacity than the polymers synthesized with one functional monomer. As presented in Figure 5B, the adsorption isotherm of CIP on MIP7 was an L-type isotherm, characteristic of a moderate affinity of adsorbent toward CIP [55]. In contrast, the adsorption isotherm on NIP7 was an S type isotherm. The maximum adsorbed amounts of CIP on MIP7 and NIP7 were 2.40 and 1.45 mg g^−1^, respectively, with the imprinting factor of 1.66. Moreover, only the imprinted polymer had a good affinity, while the non-imprinted polymer still had a weak interaction toward the target molecule. The results suggested that the hydrogen and π-π stacking interaction between MAA and 2-VP with CIP were responsible for improving the adsorption performances of the polymer.

The MIP10 and NIP10 polymers exhibited a high affinity to the CIP molecule due to the prosperous hydrophilic functional groups of MAA functional monomers. The adsorption isotherms of MIP10 and NIP10 were an H-type isotherm, characteristic of a high affinity of the matrix polymer toward CIP [55]. The maximum adsorbed amounts of CIP on MIP10 and NIP10 were 1.6 and 2.0 mg g^−1^, respectively, with the imprinting factor of 0.8. Thus, MIP10 was a non-selective adsorbent for CIP. The water-compatible property of polymers is beneficial for the access of target molecules in the adsorption process. However, the hydrogen bond between CIP and functional monomer MAA could be easily disturbed by methanol and water molecules in the synthesis protocol or the rebidding experiments. This phenomenon explained the formation of a non-selective imprinted polymer with MAA functional monomer.

The adsorption data were further analyzed using the Langmuir and Freundlich isotherm models with the non-linear method. The fitted results were summarized in Table 2. Only the adsorption isotherms of CIP on MIP10 and NIP10 were well fitted with Langmuir and Freundlich isotherm models due to their H type isotherms with high correlation coefficients (R^2^ = 0.91–0.98). In comparison, the correlation coefficients for the adsorption isotherms for other polymers (MIP2 and MIP7) were low (R^2^ = 0.44–0.87). Thus, the Langmuir and Freundlich isotherm models were unsuitable for the adsorption data on MIP2 and MIP7.

### 3.4. Solid Phase Extraction Study

#### 3.4.1. Optimization of Solid Phase Extraction Protocol

Loading condition: The loading solvent provides the appropriate environment for the adsorption of the analytes in the SPE procedure. For this study, 5 mL of CIP solution (0.1 ppm) prepared in deionized water or the mixture of phosphoric acid 0.05% (pH 3)/acetonitrile (8:2, *v*/*v*) were loaded on the cartridges. The results showed that CIP was well adsorbed onto the SPE column when deionized water was used as a loading solvent (98–100%), while only 0.1–5% CIP was retained on the cartridges when the CIP solution was prepared in the mixture of phosphoric acid 0.05 % (pH 3)/acetonitrile (8:2, *v*/*v*). Thus, ultrapure water was selected as the loading solvent for the SPE experiments.

Washing solvent: A suitable washing solvent is desired to wash out most impurities but not the analyte and improve the sensibility of the analytical method. Different washing solvents, such as dichloromethane/methanol (9:1, *v*/*v*), dichloromethane/methanol (5:5, *v*/*v*), dichloromethane/methanol (1:9, *v*/*v*), chloroform/methanol (9:1, *v*/*v*), chloroform/methanol (1:9, *v*/*v*), water, and the three steps using water followed by acetonitrile/acetic acid 0.5% (1:9, *v*/*v)* and then acetonitrile/ammoniac 0.1% (8:2, *v*/*v*) were used to optimize the washing conditions. The mixture of dichloromethane/methanol and chloroform/methanol washed out the most impurities. However, these solvents were also able to elute a large amount of the CIP retained on SPE column. Furthermore, water and the three steps with water, acetonitrile/acetic acid, and acetonitrile/ammoniac showed little influence on CIP recoveries. Therefore, water or the three steps with water, acetonitrile/acetic acid, and acetonitrile/ammoniac can be used as washing solvents, depending on the complexity of analytical samples.

Eluting solvent: The selection of elution is a critical factor to completely desorb the CIP retained on the cartridges. In this study, 5 mL of CIP aqueous solution (0.1 mg L^−1^) was firstly loaded on the SPE column. Then, 3 mL of different solutions, such as methanol/acetic acid (9:1, *v*/*v*), methanol/acetic acid (5:5, *v*/*v*), phosphoric acid 0.05% (pH 3)/acetonitrile (8:2, *v*/*v*), phosphoric acid 0.05% (pH 3)/acetonitrile (6:4, *v*/*v*), acetic acid 0.5% (pH 3)/acetonitrile (8:2, *v*/*v*), and acetic acid 0.5% (pH 3)/acetonitrile (6:4, *v*/*v*) were employed to elute the analytes. The highest CIP recovery (~100%) was observed when the solution of methanol/acetic acid (9:1, *v*/*v*) or phosphoric acid 0.05% (pH 3)/acetonitrile (8:2, *v*/*v*) was used as an eluting solvent (Figure 6). Thus, a solution of methanol/acetic acid (9:1, *v*/*v*) or phosphoric acid 0.05% (pH 3)/acetonitrile (8:2, *v*/*v*) could be used as the eluting solvent.

#### 3.4.2. Extraction Performance of Imprinted Polymer

Finally, the synthesized polymers were employed as an adsorbent for the solid phase extraction of CIP in deionized water. As shown in Figure 7, MIP2 and NIP2, which have a weak adsorption affinity, demonstrated low extraction recoveries of CIP (43–58%). The SPE cartridge with MIP7 showed an excellent recovery to CIP (105%), whereas the NIP7 cartridge showed a low recovery (43%). For non-selective polymers MIP10 and NIP10, the CIP extraction recoveries observed were identical. The low extraction recoveries observed for the SPE cartridges with these polymers (23%) could be explained by the high affinity of the MAA monomer toward CIP and that reduced the CIP elimination from the SPE column. These results indicated that the synergetic effect of dual functional monomers favored the formation of specific adsorption sites on imprinted polymers.

## 4. Conclusions

In this study, CIP-MIPs have been synthesized using MAA and 2-VP as the dual functional monomers, EDGMA as the cross-linker, and chloroform/methanol (9:1, *v*/*v*) as the solvent. The FT-IR, TGA-DSC, SEM, and nitrogen sorption isotherm characterization confirmed the physicochemical properties of the imprinted polymer, and their corresponding non-imprinted polymer was similar. The batch adsorption experiments demonstrated that the MIPs synthesized with the mixture of functional monomers (MAA and 2-VP) showed a higher adsorption capacity and selectivity toward CIP. The CIP-MIP was employed as a SPE adsorbent to extract CIP in pure water with the recovery and the relative standard deviation of 105 and 7.9%, respectively. Further studies are being performing for CIP residue analysis in the wastewater from hospitals and shrimp farms by using this CIP-MIP.

## Figures and Tables

**Figure 1 polymers-13-02788-f001:**
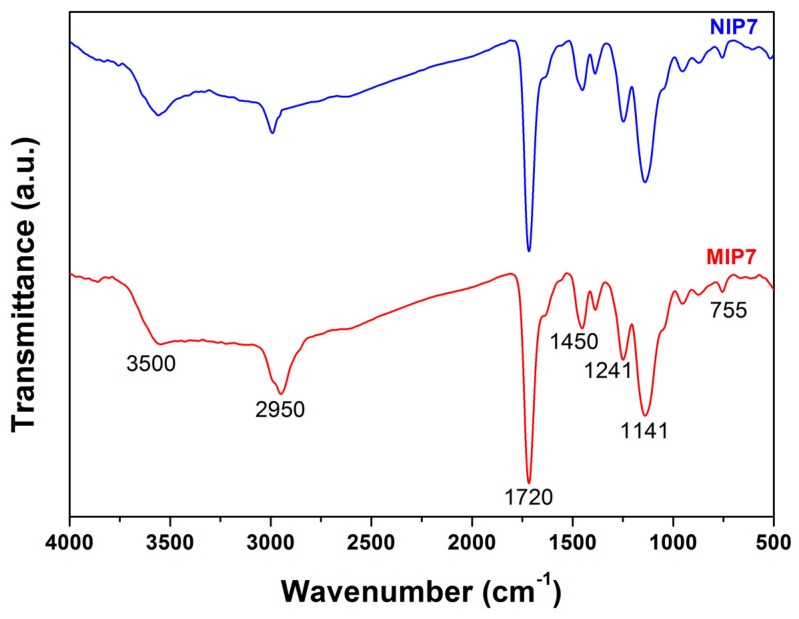
FT-IR spectra of MIP7 and NIP7.

**Figure 2 polymers-13-02788-f002:**
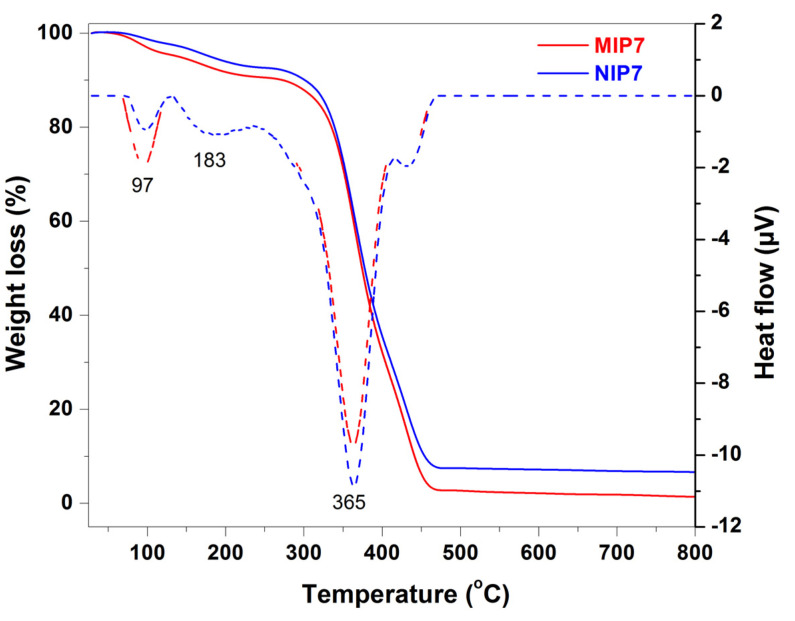
TGA analysis (solid curves) and DSC thermograms (dash curves) of MIP7 and NIP7.

**Figure 3 polymers-13-02788-f003:**
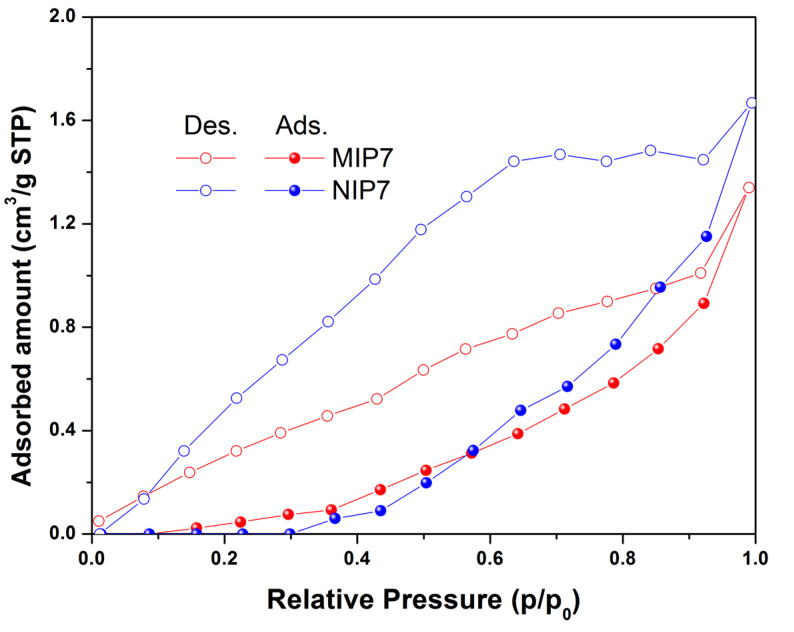
Nitrogen adsorption–desorption isotherms of MIP7 and NIP7 at 77 K.

**Figure 4 polymers-13-02788-f004:**
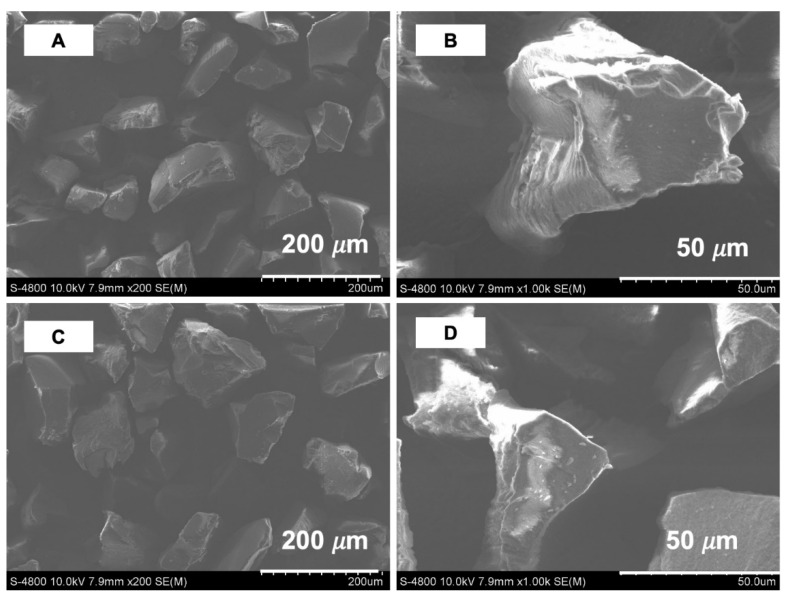
SEM images of MIP7 (**A**,**B**) and NIP7 (**C**,**D**) at different magnifications.

**Figure 5 polymers-13-02788-f005:**
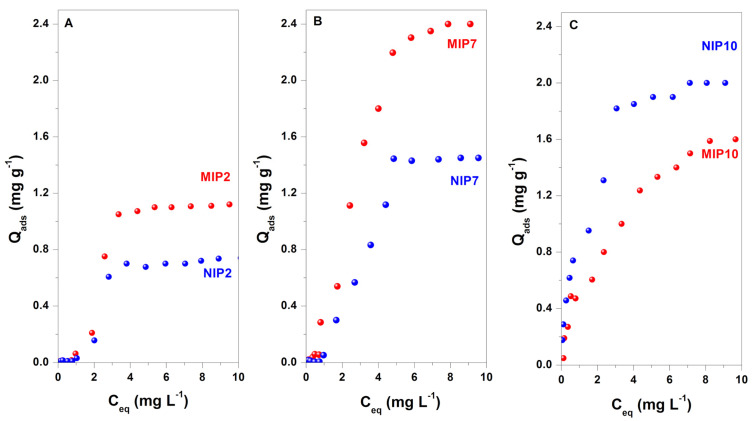
Adsorption isotherms of MIP2 (**A**), MIP7 (**B**), MIP10 (**C**), and their corresponding non-imprinted polymers.

**Figure 6 polymers-13-02788-f006:**
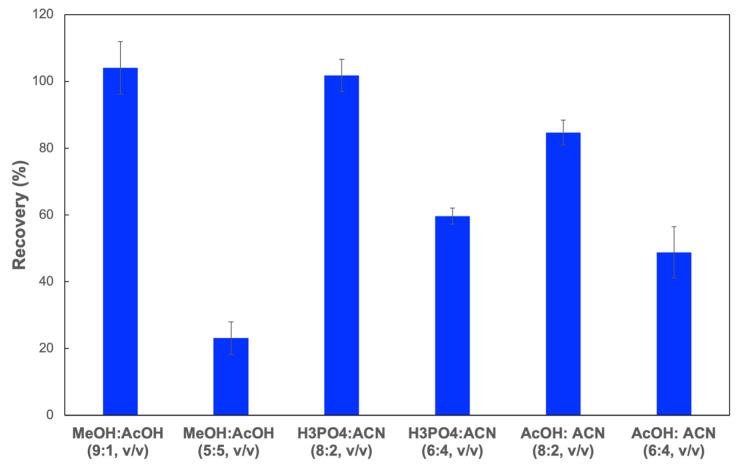
CIP recoveries from MIP-SPE column when using different eluting solvents.

**Figure 7 polymers-13-02788-f007:**
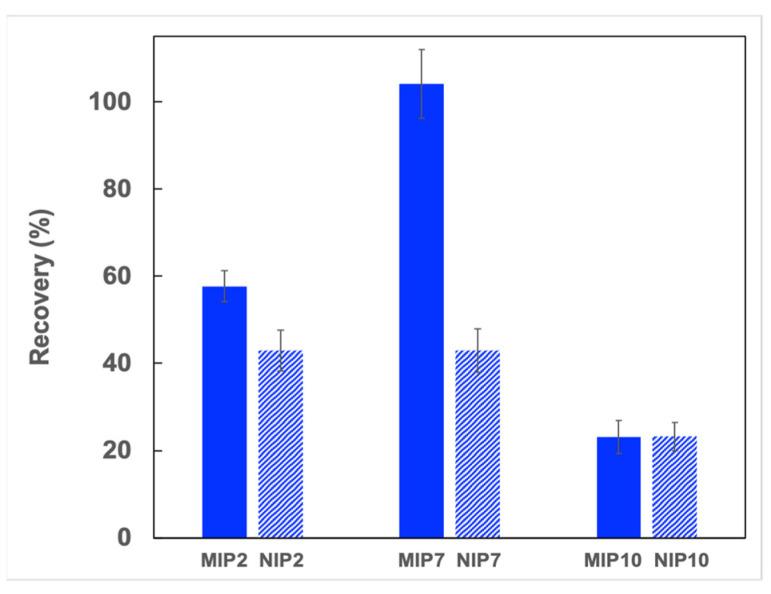
CIP recoveries from different imprinted and non-imprinted polymers (SPE condition: 40 mg of polymer, loading: 5 mL CIP solution in water (0.1 μg L^−1^), washing: 3 mL of deionized water, eluting: 3 mL of methanol/acetic acid (9:1; *v*/*v*), *n* = 3).

**Table 1 polymers-13-02788-t001:** Effect of the imprinted polymer preparation conditions on extraction recovery of CIP in water, (SPE condition: 40 mg of polymer, loading: 5 mL CIP solution in water (0.1 μg L^−1^), washing: 3 mL of deionized water, eluting: 3 mL of methanol/acetic acid (9:1; *v*/*v*), *n* = 3).

MIP	Template (mmol)	Functional Monomer (mmol)		Cross-Linker (mmol)	Porogenic Solvent	Recovery (%)	RSD (%)
		MAA	2-VP				
MIP1	1.0	-	10.0	50	MeOH: AcOH	36.5	2.7
MIP2	1.0	-	10.0	50	CHCl_3_: MeOH	57.7	5.6
MIP3	1.0	-	10.0	5.0	CH_2_C_l2_: MeOH	7.1	3.4
MIP4	1.0	-	15.0	75	CHCl_3_: MeOH	57.2	7.0
MIP5	1.0	-	20.0	100	CHCl_3_: MeOH	62.0	3.5
MIP6	1.0	5.0	5.0	50	CHCl_3_: MeOH	63.2	2.8
MIP7	1.0	6.6	3.3	50	CHCl_3_: MeOH	104.6	7.9
MIP8	1.0	7.0	3.0	50	CHCl_3_: MeOH	60.8	6.5
MIP9	1.0	8.0	2.0	50	CHCl_3_: MeOH	70.0	4.5
MIP10	1.0	10.0	-	50	CHCl_3_: MeOH	23.1	3.8

**Table 2 polymers-13-02788-t002:** Fitted results of adsorption data with Langmuir and Freundlich isotherm models using the non-linear method.

Polymer	Q_exp._ (mg g^−1^)	Langmuir			Freundlich			IF ^a^
		Q_max_ (mg g^−1^)	K_L_ (L mg^−1^)	R^2^	K_F_	n	R^2^	
MIP2	1.12	4.71	0.9353	0.4432	0.0443	0.4789	0.8117	1.51
NIP2	0.74	2.21	0.8952	0.4969	0.0270	0.5073	0.7906	
MIP7	2.40	23.92	0.9487	0.7572	0.1677	0.5873	0.8752	1.66
NIP7	1.45	34.83	0.9962	0.5297	0.0231	0.3704	0.7524	
MIP10	1.60	2.21	0.2664	0.9454	0.3691	1.3408	0.9140	0.80
NIP10	2.00	1.97	0.0754	0.9698	0.8252	2.1160	0.9883	

^a^ Imprinting factor = Q_MIP_/Q_NIP_.

## Data Availability

The data presented in this study are available on request from the corresponding author.

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
