# Peer review of "Synergetic Effect of Dual Functional Monomers in Molecularly Imprinted Polymer Preparation for Selective Solid Phase Extraction of Ciprofloxacin"

_polymers, 2021, doi:10.3390/polym13162788_

Round 1

Reviewer 1 Report

Review: polymers-1331907.

Title: Synergetic effect of dual functional monomers in molecularly imprinted polymer preparation for selective solid phase extraction of ciprofloxacin.

In this manuscript Authors present investigations of molecularly imprinted polymers (MIPs) for ciprofloxacin solid phase extraction in sample preparation treatment. The MIPs prepared from methacrylic acid, 2-vinylpyridine and ethylene glycol dimethacrylate were synthesized employing bulk polymerization process. The impact of porogen systems was evaluated. Finally, the solid phase extraction from water was carried out, revealing high efficacy of the recovery. The analysis of manuscript revealed substantial shortcomings, pointed below, that should be addressed by Authors at this stage of evaluation:

  1. The novelty of the manuscript should be clearly emphasized since there is a plethora of papers devoted to MIPs for ciprofloxacin determination even for so-called multi-functional monomer approach. Authors shall discuss convincingly the advantages of presented approach in the context of ref. 1 (Talanta 2019, 200, 307).
  2. The aspects of relative reactivity of used monomers, viz. methacrylic acid and 2-vinylpyridine should be discussed. The additional experiments that confirm the composition of polymer network shall be provided. For such purpose, the results of EDS comparative analysis and/or elemental analyses of MIP and NIP should be completed together with analysis of the FT-IR spectrum for MIP before template extraction. Moreover, the effectiveness of the template removal shall be discussed.
  3. The interactions between methacrylic acid or 2-vinylpyridine and ciprofloxacin shall be discussed more comprehensively. The interactions between monomers and the template shall be confirmed by proton NMR spectroscopy to assess the structural regions responsible for intermolecular interactions. Moreover, the molecular modeling analysis shall be also employed to explain the role of both monomers in various types of interactions with the template molecule.
  4. It is difficult to agree with the sentence provided by Authors in line 175: ‘These results indicated that the highly polar aprotic solvent (chloroform/methanol) was suitable for preparing CIP-MIP and the weakly polar solvent unfavored the interaction between the monomer and the template molecule’. Please note that chloroform/methanol system cannot be classified as the aprotic one since methanol possesses proton capable to interact.
  5. Authors should justified the selection of bulk form as most appropriate for the purpose of the solid phase extraction of ciprofloxacin. Currently, more advanced forms of MIPs are predominant as effective sorbents for extraction of analytes (see: Materials 2021, 14, 1850).
  6. In the Introduction Section the recent reviews related to MIPs for other applications should be completed (see: Chem. Rev. 2019, 119, 94, Prog. Polym. Sci. 2015, 47, 1, Trends Anal. Chem. 2021, 142, 116306, Eur. Polym. J. 2021, 143, 110179) together with recent papers dedicated to MIP for ciprofloxacin determination (see: Spectrochim. Acta 2018, 201, 382, React. Funct. Polym. 2021, 163, 104893, Sep. Sci. Technol. 2021, 56, 2217, RSC Adv. 2020, 10, 12823, Microchim. Acta 2019, 186, 334, J. Mater. Sci. 2021, 32, 1). In general the Reference Section shall be critically reviewed and updated since some of cited papers could be replaced by more actual ones.
  7. The results from nitrogen sorption analysis (BET) should be discussed in more detail way. Authors should critically discuss the impact of the porogen for the effectiveness of the MIP with respect of lesser specific surface area when compared to NIP (see: J. Sep. Sci. 2019, 42, 1412, Anal. Chim. Acta 2018, 1030, 77). The analysis of pore systems in both materials should be carried out. In my opinion, the histereses presented in Fig. 3 disclose different nature of pore systems in MIP and NIP. Moreover, the micrographs from SEM should be provided in higher magnification to disclose potential variations in the surface extension of tested materials.
  8. The data from the optimization of solid phase extraction protocol should be added. Please provide information related to the enrichment capability of the sorbent (enrichment factor).
  9. In my opinion, the lack of selectivity tests with respect to structural analogs of ciprofloxacin and other compounds that could exist in real sample is disadvantageous. The analytical methods for determination of ciprofloxacin, employing MIP systems should be discussed and properly compared with references (see references provided in entry five).
  10. In line 101, it was mentioned that ciprofloxacin was analyzed in water. What kind of sample was used, tap water, river water, sewage or just ultrapure water? If only ultrapure water was used as ciprofloxacin standard solution, it cannot reveal the impact of components of real sample on adsorption behavior of the MIP, limiting its practical application. Thus, the real sample analyses should be completed.

I hope that above mentioned suggestions  will strengthen the scientific value of the manuscript.

Therefore, in my opinion, major revision is required before final decision of the Editor.

Author Response

Dear Reviewer,

We would like to appreciate for your acceptance to review our article and provide very helpful comments to strengthen the scientific value of the manuscript. We carefully interpreted all comments and tried our best to provide additional data and improve our paper. We hope that the revised manuscript will reach the quality of your expectations and consideration for publication in Polymers. In addition, we are willing to replay if you have any other comments.

Please find enclosed the response point by point to your comments. 

Thanks for your contribution.

Sincerely,

Dr. Ut Dong Thach.

Reviewer 2 Report

It is a rather good article

I do not understand this sentence: Please reformulate:

When the  template molecules containing polar groups, the selectivity of synthesized MIP is impaired by water molecules1,31,3.

You have also some small english mistakes as for example:

The band at 755 cm–1 was characteristic for C-H out-of-plant bending vibration

instead of:

The band at 755 cm–1 was characteristic for C-H out-of-plan bending vibration

or:

The results of textural properties analyzed by nitrogen adsorption-desorption isotherm shown that the synthesized polymers….

instead of:

The results of textural properties analyzed by nitrogen adsorption-desorption  isotherm have shown that the synthesized polymers….

Please read again the manuscript and make corrections.

I do not think that:

"The final endothermic peak at 431 °C  could be explained by the decomposition of the carbon backbone"

I think instead that this peak is explained by the pyrolysis with carbonization of the backbone.

Can you give some literature about this?

You say that:

selective factor KMIP/NIP of 1.51.

The ratio between MIP and NIP absorption is named in the literature Imprinting factor (IF) not selective factor. Selective factor refers in the literature at the ratio of absorption of a MiP for the target and for another  concurrent (usually with a rather similar structure). So that , please change KMip/NIP with IF. Change in all manusript selective factor with imprinting factor!

Please change the phrase, which is in poor English.

Noticed that the MIP synthesized with dual functional monomers showed higher adsorption capacity than MIP synthesized using individual functional monomers

I do not see that:

As presented in Figure 5B, the adsorption isotherm of CIP on MIP7 was L-type isotherm, characteristic of a moderate affinity of adsorbent towards CIP. In contrast, the adsorption  isotherm on NIP7 was S type isotherm.

nor

The adsorption  isotherms of MIP10 and NIP10 were an H-type isotherm!

it is better to use IUPAC type: I, II, III, IV

Author Response

Dear Reviewer,

We would like to appreciate your acceptance to review our article and provide very helpful comments to strengthen the scientific value of the manuscript. We carefully interpreted all comments and tried our best to improve our paper. We hope that the revised manuscript will reach the quality of your expectations and consideration for publication in Polymers. In addition, we are willing to replay if you have any other comments.

Please find enclosed the response point by point to your comments.  

Thanks for your contribution.

Sincerely,

Ut Dong Thach.

Round 2

Reviewer 1 Report

Review: polymers-1331907_R1.

Title: Synergetic effect of dual functional monomers in molecularly imprinted polymer preparation for selective solid phase extraction of ciprofloxacin.

In this revised manuscript, Authors have made corrections according to referee comments. Although not all suggestions were implemented into the text and some parts of the manuscript still could be improved, I propose consideration of acceptance.

Author Response

Dear reviewer,

I appreciate your consideration.

Sincerely,

Dr. Ut Dong Thach.

Reviewer 2 Report

 The article is improved. However there are still some problems, especially concerning English quality.

I do not understand this phrase:

in which the conventional bulk imprinting technique has been still em ployed to synthesize imprinted material due to it simple, rapid, and pure MIPs production can be produced without sophisticated instrumentation.

The phrase bellow is in poor English

About 10 mg of the polymer was equilibrating with 20 mL CIP solution in a 50 mL centrifuge tube.

I do not understand this:

As shown in Table 1, the recovery was statistically comparable with the number of functional monomers.

Change:

The characteristics absorption peaks of 2-VP appeared at 1637 and 1450 cm–1 were due to the C=N and C=C…

With:

The characteristics absorption peaks of 2-VP appearing at 1637 and 1450 cm–1 were due to the C=N and C=C…

Change:

The corresponding DSC thermogram of imprinted polymer (MIP7) showed two endothermic peaks at 97 and 183 °C were due to removing adsorbed water and entrapped solvent molecules.

With:

The corresponding DSC thermogram of imprinted polymer (MIP7) showed two endothermic peaks at 97 and 183 °C, which were due to removing of the adsorbed water and of the entrapped solvent molecules, respectively.

In figure 2 show which curves refers to a certain ordinate.

Change:

Nearly a total of CIP (98-100%) was adsorbed onto  SPE column when deionized water was used as loading solvent. While only 0.1-5% CIP  was retained on the cartridges when the CIP solution was prepared in the mixture of phosphoric acid 0.05 % (pH 3)/acetonitrile (8:2, v/v).

With:

Nearly a total of CIP (98-100%) was adsorbed onto  SPE column when deionized water was used as loading solvent,  while only 0.1-5% CIP  was retained on the cartridges when the CIP solution was prepared in the mixture of phosphoric acid 0.05 % (pH 3)/acetonitrile (8:2, v/v).

Put a reference in text to figure 6. Change the figure caption : It is figure 6 not figure 1

Author Response

Dear Reviewer,

Thank you for your corrections concerning English quality. We have made all corrections in the revised version.

Sincerely,

Ut Dong Thach.
